# Effect of Wheat-Derived Arabinoxylan on the Gut Microbiota Composition and Colonic Regulatory T Cells

**DOI:** 10.3390/molecules28073079

**Published:** 2023-03-30

**Authors:** Seita Chudan, Riko Ishibashi, Miyu Nishikawa, Yoshiaki Tabuchi, Yoshinori Nagai, Shinichi Ikushiro, Yukihiro Furusawa

**Affiliations:** 1Department of Biotechnology, Faculty of Engineering, Toyama Prefectural University, Toyama 939-0398, Japan; 2Department of Pharmaceutical Engineering, Faculty of Engineering, Toyama Prefectural University, Toyama 939-0398, Japan; 3Division of Molecular Genetics Research, Life Science Research Center, University of Toyama, Toyama 930-0194, Japan

**Keywords:** arabinoxylan, wheat, gut bacteria, regulatory T cells, colitis, short-chain fatty acid

## Abstract

The health benefits of wheat-derived arabinoxylan, a commonly consumed dietary fiber, have been studied for decades. However, its effect on the gut microenvironment and inflammatory bowel disease remains unclear. The objective of this study was to understand the effect of wheat-derived arabinoxylan on gut microbiota, colonic regulatory T cells (Tregs), and experimental colitis. In this study, healthy and chronic colitis model mice were fed chow containing cellulose or wheat-derived arabinoxylan for 2–6 weeks and subjected to subsequent analysis. A 16S-based metagenomic analysis of the fecal DNA revealed that Lachnospiraceae, comprising butyrate-producing and Treg-inducing bacteria, were overrepresented in arabinoxylan-fed mice. In line with the changes in the gut microbiota, both the fecal butyrate concentration and the colonic Treg population were elevated in the arabinoxylan-fed mice. In a T cell transfer model of chronic colitis, wheat-derived arabinoxylan ameliorated body weight loss and colonic tissue inflammation, which may, in part, be mediated by Treg induction. Moreover, wheat-derived arabinoxylan suppressed TNFα production from type 1 helper T cells in this colitis model. In conclusion, wheat-derived arabinoxylans, by altering the gut microenvironment, may be a promising prebiotic for the prevention of colitis.

## 1. Introduction

Dietary fiber shortage is a recognized risk factor for the onset of metabolic and immune diseases. Among immune disorders, inflammatory bowel disease (IBD), classified into ulcerative colitis and Crohn’s disease, is attributed to lifestyle changes, particularly a diet characterized by low fiber content that modifies the intestinal microenvironment [1]. Therefore, the current clinical practice guidelines for patients with IBD encourage dietary fiber supplementation during IBD remission to prevent relapse [2].

Dietary fiber is a mixture of polysaccharides including 10 or more monosaccharides and is not hydrolyzed by human gastrointestinal enzymes [3]. Dietary fiber is consumed by commensal bacteria abundantly present in the large intestine and is thus an energy source for these bacteria, resulting in a change in gut bacterial composition [4]. In this process, the gut bacteria ferment the dietary fiber and produce metabolites, such as short-chain fatty acids (SCFAs), which modulate the immune function of both lymphocytes and myeloid cells and are an energy source for colonic epithelial cells [5,6]. Among the SCFAs, accumulating evidence has demonstrated that butyrate maintains the gut mucosal immune system by promoting regulatory T cell (Treg) differentiation as well as a T-cell-independent IgA response through histone deacetylase inhibition and G-protein-coupled receptor (GPR) activations [7,8,9,10].

The digestibility and fermentation rates of dietary fibers are dependent on their molecular weight, viscosity, hydrophilicity, degree of branching, monosaccharide content, and linkage composition [11]. Among the many kinds of dietary fibers, arabinoxylan (AX) derived from wheat is known to be a highly fermentable fiber owing to its relatively high degree of arabinose substitution on the xylan backbone [12]. In general, AX consists of a linear β(1–4)-linked xylan backbone to which a-1-arabinofuranose units are attached as side residues via α-(1–3) and/or α (1–2) linkage. Xyloses are most commonly mono-substituted, but the degree of substitution is influenced by the wheat variety and the wheat grain’s maturation. Compared with rice bran AX, wheat AX has a less branched structure, and (1–5)-linked arabinose is absent (reviewed in [13]). Wheat AX fermentation takes place in the transverse to distal colon locations rather than the proximal colon, resulting in slow fermentation and effective alteration in the gut microenvironment [14]. In contrast, fermentation of cellulose (CE), the most abundant dietary fiber in plants, is minimal in the human gut [15].

Recently, whole grains, such as wheat bran, have attracted increasing attention as a source of dietary fiber. Wheat bran accounts for 30–50% of dietary fiber and consists of approximately 64–69% of AX and 15–31% of CE as the major non-starch polysaccharides [16]. The beneficial effects of wheat bran and AX on the gut microenvironment and health have been extensively studied. Balb/c mice fed wheat bran showed increased luminal IgA production, which coincided with Clostridiales overrepresentation and elevated SCFA levels [17,18]. This IgA induction may be attributed to the AX from the wheat bran because soluble AX feeding also augmented IgA production in the Balb/c mice [19]. Elevated fecal SCFAs, particularly propionate and butyrate, due to wheat bran intervention were recently confirmed even in healthy human adults, although the change in gut microbiota composition was dependent on individual characteristics, perhaps their different enterotypes that contribute to different lifestyle habits, as well as genetic background [20]. To minimize the differences between individuals during interventions, Paepe et al. evaluated the effect of wheat bran on gut microbiota composition and fermentable properties using a Human Intestinal Microbial Ecosystem model [21]. This model demonstrated that wheat bran augmented *Lactobacillus* spp. and several SCFA-producing bacteria (e.g., Roseburia, Prevotella, and Bacteroides), which coincided with elevated propionate and butyrate concentrations [21]. These results indicated that AX from wheat bran may promote SCFA production by altering the composition of the gut microbiota, even in humans.

Several studies have evaluated the effects of wheat bran and wheat-derived AX on gut bacteria, SCFA production, and IgA secretion. However, few studies have demonstrated the beneficial effects of AX in IBD. Only one recent study has evaluated the impact of AX intervention on DSS-induced colitis, an IBD model triggered by an intestinal barrier disorder, followed by bacterial invasion and subsequent innate immune cell activation [22]. However, no conclusions have been drawn regarding the preventive effects of AX against colitis. Due to a lack of evidence, the effect of wheat AX on immunological function and colitis, particularly in cases triggered by the acquired immune system, remains to be elucidated.

In the present study, we evaluated the beneficial effects of wheat-derived AX on T-cell-dependent colitis in terms of changes in the gut microenvironment. We managed to demonstrate that wheat-derived AX induces colonic peripherally induced Treg (pTreg) cells and ameliorated T-cell-dependent chronic colitis, which may be a result of increased butyrate-producing bacteria and luminal butyrate levels.

## 2. Results

### 2.1. Arabinoxylan Alters Gut Microbiota Composition with Little Effect on the Number of Species in Mice

To determine the impact of AX on the gut bacterial composition, we collected feces from mice fed chow containing CE or wheat-derived AX (AX) for 2 weeks, unless otherwise specified, followed by 16S rRNA-based metagenome sequencing. In this experiment, we chose AIN-76A as the basal chow because the general rodent diet (e.g., CE-2) includes wheat and rice bran, which contains AX as a source of crude fiber.

Firstly, we analyzed α- and β-diversity to compare the number of observed species and the microbial composition in the CE- and AX-fed mice. The α-diversity (rarefaction) analysis showed a slight, but not significant, increase in the number of microbial species in the mice fed AX (Figure 1A). On the contrary, the β-diversity analysis (principal component analysis) revealed that the bacterial composition patterns were markedly distinct between the CE- and AX-fed mice (Figure 1B), indicating that the bacterial composition, rather than the number of species, was altered by AX-feeding. Next, to identify the altered gut microbiota in the AX-fed mice, we performed a taxonomic classification of 16S rRNA-targeted amplicons. In the taxonomic analysis at the phylum level, Firmicutes, Bacteroidetes, and Proteobacteria were the dominant communities in both groups. Among these phyla, Firmicutes were overrepresented, whereas Bacteroidetes were underrepresented in the AX-fed mice (Figure 1C). Proteobacteria, a major phylum of Gram-negative bacteria comprising several pathogenic bacteria [23], were less abundant in the AX-fed mice (Figure 1C).

### 2.2. Bacterial Taxa Altered by Arabinoxylan Intervention

In the taxonomic analysis at a family–species levels, Lactobacillus spp. and Lachnospiraceae were the predominant Firmicutes bacteria in the mouse feces (Figure 2A). The abundance of these bacteria in the AX-fed mice was more than those in the CE-fed mice (Figure 2A). The LEfSe analysis, which is used for identifying multilevel specie differences, also represented differences of Lactobacillus spp. and Lachnospiraceae between the CE- and AX-fed mice with a high linear discriminant analysis (LDA) score > |4.0| (Figure 2B,C). As for the bacteria belonging to Bacteroidetes, Bacteroides AB599946, Alloprevotella PAC002479, and unclassified Bacteroides spp. were abundant in the mouse feces (Figure 2A). Among them, unclassified Bacteroides spp. were overrepresented but the others were underrepresented in the AX group (Figure 2A,C). A cladogram also showed that some Bacteroides spp. (labeled as “c”, “g”, “i”) and Parabacteroides spp. (labeled as “j”), but not others, were abundant in the AX-fed mice (Figure 2B). Regarding the bacteria classified into proteobacteria, Desulfovibrionaceae, a sulfate-producing bacteria abundant in patients with ulcerative colitis [24], were slightly underrepresented in the AX group (Figure 2A,C).

### 2.3. Arabinoxylan Augments Fecal Propionate and Butyrate

Altered gut microbiota composition by AX may affect the concentration of colonic metabolites such as SCFAs. To assess the effect of AX on SCFA production in the gut, we performed gas chromatography (GC) analysis and quantified the SCFA concentrations in the feces collected from the mice fed CE or AX for 2 weeks. In support of the elevated levels of Lachnospiriaceae, which are butyrate-producing bacteria [25], the fecal butyrate levels were significantly elevated in the mice fed AX (Figure 3). Some Bacteroides spp. produce propionate via the succinate pathway [5]. The fecal propionate levels were also elevated in the AX-fed mice, although the response to AX varied among the Bacteroides species (Figure 2 and Figure 3). In contrast, fecal acetate levels were only slightly increased in the AX-fed mice, but statistical significance was absent between the two groups, which may be due to the higher basal concentration of acetate than that of propionate and butyrate (Figure 3).

### 2.4. Arabinoxylan Augments Colonic Peripherally Induced Treg Cells

Lachnospiraceae, also known as Clostridium cluster XIVa, comprises Treg-inducing bacteria [26]. In addition, intestinal butyrate markedly promoted colonic Treg differentiation [7]. To assess the Treg-inducing ability of AX, we analyzed the colonic lamina propria Tregs in the mice fed CE or AX for 3 weeks. Consistent with the elevated Lachnorpiraceae abundance and butyrate levels, the population of colonic pTreg cells (Foxp3^+^RORγt^+^Helios^−^ CD4^+^ T cells) was also significantly increased in the AX-fed mice (Figure 4A,B). In contrast, the percentage of thymus-derived Treg (tTreg: Foxp3^+^RORγt^−^Helios^+^ CD4^+^ T cells) cells was comparable between the CE and AX groups (Figure 4C), indicating that AX promoted colonic pTreg differentiation or proliferation rather than migration.

### 2.5. Arabinoxylan Ameliorates T-Cell-Dependent Chronic Colitis

Next, we investigated the preventive effects of AX on T-cell-dependent inflammation by establishing a T cell transfer model of chronic colitis. Naïve CD4+ T cells were transferred into Rag1^−/−^ mice, which were then fed either CE or AX for 6 weeks. AX recovered body weight loss and the shortening of colon length, which are typical indicators of the severity of colonic inflammation (Figure 5A,B). The pathological changes indicated by leukocyte infiltration and loss of goblet cells in the AX-fed mice were less severe than those in the CE-fed mice (Figure 5C,D).

To further investigate the underlying mechanism by which AX alleviates colonic inflammation, we analyzed the colonic T cell population and phenotype in these mice. Consistent with the results shown in Figure 3, the percentage of colonic Tregs was also increased by AX feeding, even in the colitis model, although the relative Treg population in CD4^+^ T cells in the adoptive transfer model was lower than that in the wild-type healthy mice (Figure 6A,B and Figure 4A). In contrast, more than 50% of the CD4^+^ T cells showed a type 1 helper cell (Th1) phenotype that produced IFNγ and TNFα (Figure 6C). In this Th1-prone model, AX intervention significantly decreased IFNγ/TNFα double-positive cells but not IFNγ or TNFα single-positive cells compared to the CE group (Figure 6C–F).

## 3. Discussion

The findings of the present study demonstrate that wheat-derived AX induces colonic pTregs and alleviates T-cell-dependent chronic colitis, which may be attributed to changes in the gut microbiota and elevated SCFA levels, particularly butyrate. Butyrate is produced by bacterial species in the Lachnospiraceae family (e.g., *Eubacterium*, *Roseburia* spp.) through the butyryl-CoA:acetate CoA-transferase pathway [27]. The molecular mechanisms underlying the anti-inflammatory effects of butyrate have been extensively studied over the last decade. For example, our group and others have demonstrated that intestinal butyrate produced by dietary fiber fermentation promotes Treg differentiation through HDAC inhibition in naïve T cells and GPR109a activation in dendritic cells [7,9]. HDAC inhibition and GPR109a activation by butyrate also polarize macrophages to the anti-inflammatory M2 phenotype [9,28]. Butyrate also promotes IL-10 production from intestinal epithelial cells in a GPR109a-dependent manner [6], indicating that the preventive effect of AX on chronic colitis may be explained by the butyrate-dependent functional modification of the colonic tissue cells.

Although not as much as butyrate, propionate induces Tregs both in vivo and in vitro. The underlying mechanism by which propionate induces Treg differentiation appears to be via HDAC inhibition rather than activation of its receptor (e.g., GPR41 and GPR43 [29]) because acetate, which has a much lower HDAC inhibitory effect but a high affinity for these GPRs [29], does not affect Treg induction both in vitro and in vivo [7] (except in the presence of non-physiological high concentrations in the blood [30]). Propionate is produced by bacteria, such as *Bacteroides* spp. and *Veillonella* spp., via the succinate pathway [27]. In this experiment, AX augmented propionate production, which might have been a result of the fermentable properties of AX as well as the overrepresentation of certain *Bacteroides* spp. Considering that propionate also has the potential to increase colonic Tregs, this may also explain a part of the mechanism underlying the anti-inflammatory properties of AX.

In this study, AX suppressed TNFα production from the IFNγ-, producing Th1 cells (Figure 6C–F). Supporting this result, AX intervention reduced TNFα production from stimulated peripheral blood mononuclear cells isolated from humans [31]. A previous study demonstrated that butyrate inhibits the NF-kappa B signaling pathway in LPS-stimulated RAW264.7 cells and ameliorates colitis in *Il10*^−/−^ mice [32], indicating the possibility that butyrate derived from AX might directly suppress TNFα production from Th1 cells as well as macrophages. An in vitro study also reported that butyrate treatment augmented IL-10 production from Th1-polarized cells without affecting the percentage of IFN-producing cells [33]. Although the direct effect of butyrate on the immunological function of Th1 cells remains to be elucidated, butyrate derived from AX ameliorates colitis, possibly through the direct modification of Th1 function and Treg induction.

AX may even show anti-inflammatory effects in intestinal inflammation following an infection. A previous study showed that wheat bran ameliorates infectious enteritis by modifying gut bacteria and SCFA production [34]. In their study, wheat bran counteracted inflammatory cytokine expression (e.g., TNFα) that was induced by *Citrobacter rodentium* infection, which coincided with the overrepresentation of butyrate-producing bacteria (*Dorea*, *Ruminococcus*, and *Roseburia* spp.). Both our study and their study commonly presented an overrepresentation of butyrate-producing bacteria and elevated butyrate levels by diet intervention, supporting the idea that the anti-inflammatory effect of AX is attributed to changes in the gut microenvironment.

In contrast, a previous study failed to determine a preventive effect against DSS-induced acute colitis, which is triggered by gut barrier destruction and subsequent innate immune cell activation [22]. The reason that AX did not affect DSS-induced colitis might be due to the low dosage of AX used in the previous report. The 200 mg/kg dose for two weeks used in the report may be insufficient to elevate colonic SCFA levels, although they did not quantify the luminal SCFA concentration. A previous study demonstrating the beneficial effects of wheat bran on IgA production used a high dose (~2500 mg/kg for 4 weeks, equivalent to ~750 mg AX/kg) [19]. Another study demonstrated that elevated SCFA levels and IgA production substituted equal amounts of wheat-bran-derived fiber with 5% cellulose in the AIN-93G diet [17], which may be calculated as approximately 1200 mg/kg daily intake of AX per mouse. A high dose of AX elevated the propionate and butyrate levels, even 1 week after AX feeding, which was maintained until the end of the experiment (at 4 weeks). Considering that butyrate has the potential to strengthen intestinal barrier functions and suppress inflammatory responses, even in innate immune cells [5], AX may alleviate DSS-induced acute colitis if the dosage (amount and duration) is sufficient to elevate the luminal SCFAs. Details regarding the anti-inflammatory effects of AX in other colitis models such as the DSS-induced colitis model remain to be elucidated.

In conclusion, this is the first study to demonstrate the beneficial effects of AX in a mouse model of T-cell-dependent chronic colitis, which may result from changes in the gut microenvironment. Although the anti-inflammatory effects of AX require further elucidation, we propose that AX may be a promising prebiotic candidate for IBD prevention.

## 4. Materials and Methods

### 4.1. Animals and Dietary Fibers

C57BL/6J male mice (5 weeks old) were purchased from Japan SLC (Hamamatsu, Japan) and maintained under standard temperature, humidity, and light conditions (20–25 °C, 50 ± 10%, 12 h light/dark cycle, respectively). The mice were fed AIN-93G (Research Diet, New Brunswick, NJ, USA) for an acclimation period of 1 week, and AIN-76A, in which corn starch was converted to dextrose (Research Diet, New Brunswick, NJ, USA), to minimize the influence of resistant starch on the gut microenvironment [35]. In the experiments, 5% of the cellulose in the modified AIN-76A was replaced with 5% AX. Naxus (Bioactor, Maastricht, Netherlands) was used as the source of AX following previous studies [12,36,37]. The estimated concentration of AX in mg/kg food and that in mg/kg body weight per day was 50 mg/kg and 600 mg/kg, respectively. The duration of AX feeding was 2, 3, and 6 weeks as shown in Figure 1, Figure 2, Figure 3, Figure 4, Figure 5 and Figure 6, respectively.

### 4.2. Gas Chromatography (GC) Analysis of Cecal and Fecal SCFAs

Fecal SCFAs were measured using GC as previously described [38]. Briefly, the cecal contents and feces were placed in 10 volumes of water and homogenized using a toothpick to extract the cecal and fecal acetate, propionate, butyrate, and valerate. The supernatants were transferred to new tubes and heptanoic acid was added as an internal standard. For derivatization, boiling stones, NaOH, pyridine, and isobutanol were added to the supernatant according to the protocol provided by Agilent Technologies (Santa Clara, CA, USA). The derivative was extracted using hexane and subjected to GC using an Agilent 7820A instrument equipped with a flame ionization detector and a DB-WAX Ultra Inert column (30 m × 250 μm × 0.5 μm). Helium was used as the carrier gas at a flow rate of 1 mL/min. The initial oven temperature of 40 °C was maintained for 5 min and then increased to 250 °C at a rate of 10 °C/min. The injected volume was 1 μL and the split ratio was 5:1.

### 4.3. Isolation of Fecal DNA

Fecal samples were collected from the mice who were fed the AIN-76A-based modified diet for two weeks. DNA was extracted from ≤100 mg of feces using the ZymoBIOMICS DNA Miniprep kit (Zymo Research, Irvine, CA, USA), according to the manufacturer’s protocol. A Disruptor Genie instrument (Scientific Industries, Bohemia, NY, USA) was used for bead beating, for 20 min, at maximum speed.

### 4.4. 16S ribosomal RNA (rRNA) Sequencing

The 16S rRNA sequencing was performed using the MiSeq Reagent Kit v2 according to the guidelines provided by Illumina (San Diego, CA, USA). The 16S rRNA reads were analyzed using Quantitative Insight into Microbial Ecology 2 (QIIME2) Ver.2021.2. A Cutadapt plugin was used to trim the primer region (forward: 17 bases; reverse: 21 bases) from the raw sequences, followed by joining of paired-end reads (forward: 250 bp; reverse: 250 bp). Amplicon sequence variants (ASV) were constructed from the processed reads using the DADA2 algorithm. To perform α- and β-diversity analyses, a diversity core-metrics-phylogenetic analysis was used for 10,000 reads set at the sampling depth. Furthermore, the feature classifiers, classify-sklearn and EZBiocloud databases [39], were used to assign the taxonomy. The microbiota features were characterized using linear discriminant analysis (LDA) and size effect (LEfSe) analysis [40], which emphasize the statistical significance and biological relevance. An effect size threshold of 2.0 and a *p*-value threshold of 0.05 were used as the cut-off values.

### 4.5. Preparation of Colonic Lymphocytes

Colonic lamina propria (LP) cells were isolated as previously reported [7,41]. Briefly, the colonic tissues were treated with Hank’s balanced salt solution containing 20 mM EDTA and 1 mM DTT at 37 °C for 20 min to remove the epithelial cells. The tissues were then minced and dissociated with the RPMI 1640 medium containing 0.5 mg/mL collagenase (Wako Pure Chemical Industries, Wako, Tokyo, Japan) and 0.5 mg/mL DNase (Roche, Basel, Switzerland), at 37 °C, for 20–30 min, to obtain a single-cell suspension. The suspension was filtered, washed with RPMI 1640, and separated using a 40/80% Percoll gradient.

### 4.6. Flow Cytometry

Staining was performed as previously described [7,38,41]. For intracellular staining of Foxp3, RORγt, and Helios, the isolated cells were stained with monoclonal antibodies (mAbs) against CD3e and CD4 (both from BioLegend, San Diego, CA, USA) and subsequently fixed with a True-Nuclear Transcription Factor Buffer Set (BioLegend). The cells were then stained with mAbs against Foxp3 (Thermo Fisher Scientific, Waltham, MA, USA), RORγt (BD Bioscience, Franklin Lakes, NJ, USA), and Helios (BioLegend), and then subjected to flow cytometry. For the intracellular cytokine staining, the LP cells were cultured for 5.5 h in a complete medium supplemented with 50 ng/mL PMA, 500 ng/mL ionomycin, and 5 μg/mL brefeldin A (Sigma, Burlington, MA, USA). The cells were harvested and stained with mAbs against cell surface antigens, followed by fixation in a fixation/permeabilization buffer (BioLegend). The fixed cells were subjected to intracellular staining, with mAbs, against IFNγ and TNFα (BioLegend). The stained samples were analyzed using a NovoCyte Flow Cytometer (ACEA Biosciences, San Diego, CA, USA) and the FlowJo software version 10 (TOMY Digital Biology, Tokyo, Japan).

### 4.7. Induction of Colitis by Adoptive Transfer of the Naïve CD4^+^ T Cell

Colitis was induced in the *Rag1*^−/−^ mice by adoptive transfer of naïve CD4^+^ T cells, as described previously, with a few modifications [7]. Briefly, the naïve CD4^+^ T cells were enriched from the splenocytes of C57BL/6J mice using an Easysep Naïve CD4^+^ T cell isolation kit (Veritas, Santa Clara, CA, USA). The *Rag1*^−/−^ recipients were administered 2 × 10^5^ naïve CD4^+^ T cells via the peritoneum and sacrificed 6 weeks after transfer.

### 4.8. Histology

Colonic tissue samples were fixed overnight in MildForm 10N (Wako). Following fixation, the samples were embedded in paraffin and cut into 3 μm sections. The sections were deparaffinized, rehydrated, and stained with hematoxylin and eosin (H&E), Alcian blue, and periodic acid–Schiff (PAS). The histological score of the adoptive transfer colitis model was evaluated on a 3-point (score 1–3) scale for leukocyte infiltration and goblet cell loss (minimum: 2 and maximum: 6 in total).

### 4.9. Statistical Analysis

Values are expressed as mean ± standard deviation (SD). The Student’s t-test or Welch’s *t*-test were used for comparisons between the two groups. R version 3.2.1, or Statistical Analysis for Mac version 3.0, were used for the statistical analyses. For multiple comparisons of the 16S rRNA sequencing data, statistical values were calculated using Welch’s t-test and Bonferroni correction, followed by Benjamini and Hochberg’s false discovery rate correction. *p* < 0.05 was used as the threshold for statistical significance.

## Figures and Tables

**Figure 1 molecules-28-03079-f001:**
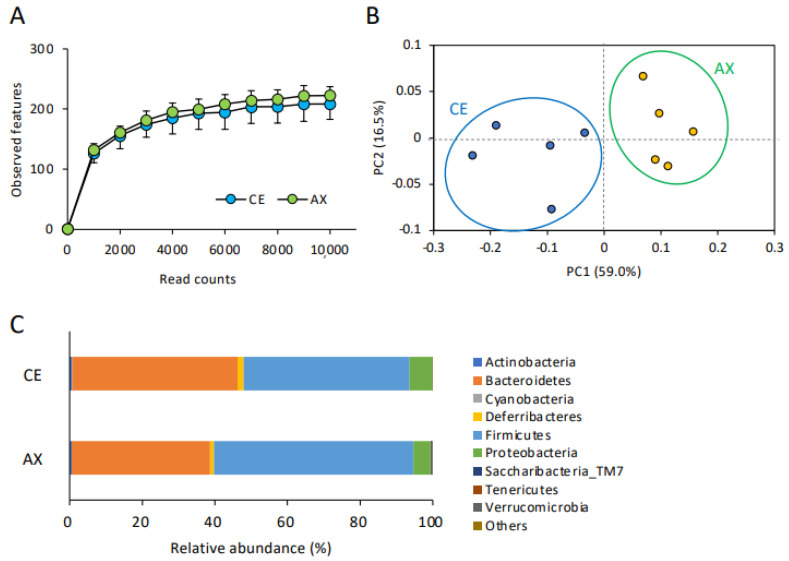
Effect of AX on the gut microbial diversity and composition: (**A**) The impact of AX on the α-diversity (observed features) of the gut bacteria analyzed by 16S rRNA sequencing (n = 5). Values and error bars indicate the mean ± SD. (**B**) Principal component analysis of the β-diversity values of the gut bacteria. (**C**) The effects of AX on the phylum-level gut bacterial composition.

**Figure 2 molecules-28-03079-f002:**
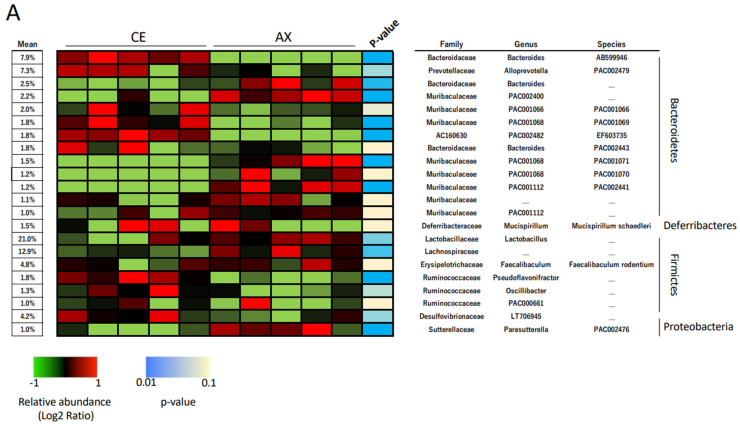
Relative abundance of the gut bacteria at the family, genus, and species levels: (**A**) The effects of AX on the relative abundance of shared bacterial taxa (n = 5). Bacterial taxa with a mean relative abundance greater than 1% are shown in the heatmap. Blue and light blue represent a statistical significance between the CE and AX groups; yellow indicates no statistical difference. (**B**,**C**) Graphics of linear discriminant analysis (LDA) effect size (LEfSe) for the bacterial taxa responsive to AX. A cladogram (**B**) and horizontal bar plot (**C**) depicting the results of the LEfSe analysis for each taxon. The length of the bar represents the log10-transformed LDA score. Bacterial taxa with a mean relative abundance greater than 1% were subjected to LEfSe analysis. Bacterial taxa abundant in the feces of the AX-fed mice are shown in red and those of the CE-fed mice are shown in green.

**Figure 3 molecules-28-03079-f003:**
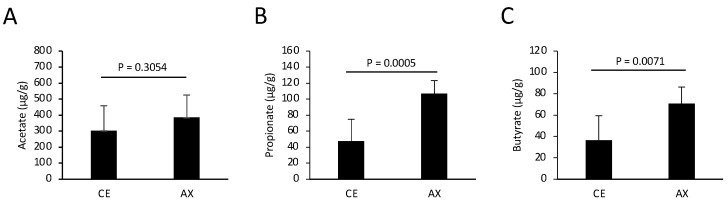
Quantification of fecal SCFAs in the mice fed chows containing CE or AX (n = 8). The feces were collected 2 weeks after feeding and subjected to GC analysis. (**A**) Acetate. (**B**) Propionate. (**C**) Butyrate. Values and error bars indicate the mean ± SD.

**Figure 4 molecules-28-03079-f004:**
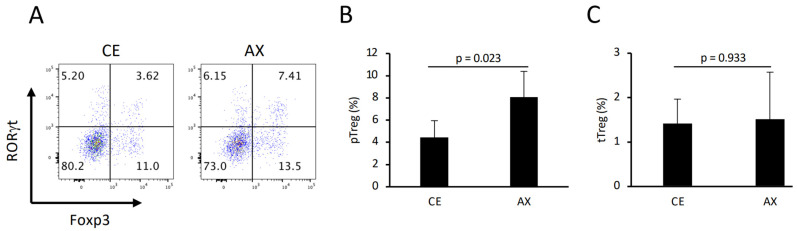
The percentage of colonic pTreg and thymus-derived Treg (tTreg) cells in mice fed CE or AX: (**A**) Typical FACS panel showing Foxp3 and RORγt expression in colonic CD4^+^ T cells. (**B**,**C**) Percentage of pTreg and tTreg cells, defined as Foxp3^+^RORγt^+^Helios^-^ or Foxp3^+^RORγt^−^Helios^+^ cells, respectively. (n = 6). Values and error bars indicate the mean ± SD.

**Figure 5 molecules-28-03079-f005:**
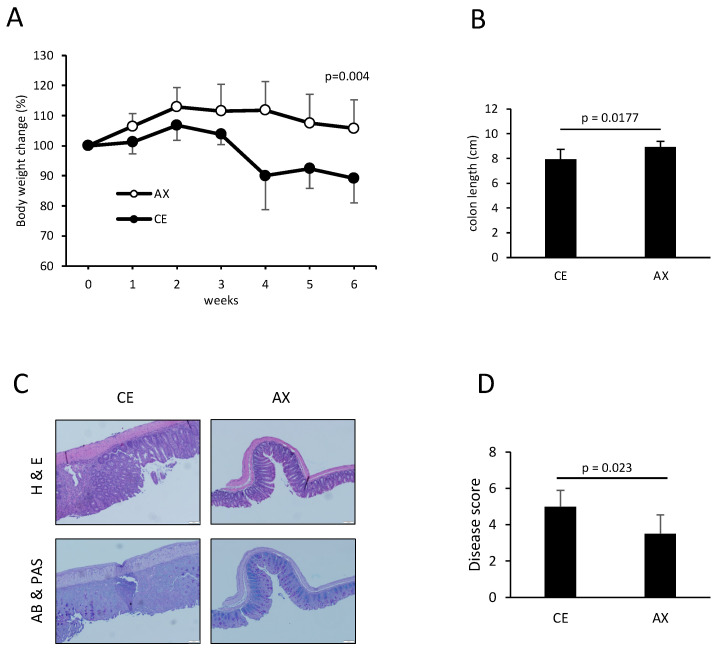
The effects of AX on T-cell-dependent chronic colitis: (**A**) Body weight changes in the mice after the adoptive transfer of naïve CD4^+^ T cells in the Rag1^−/−^ mice (n = 7). (**B**) Colon length 6 weeks after the adoptive transfer of naïve CD4^+^ T cells (n = 7). (**C**,**D**) Histology and disease score of the colonic tissues (n = 6). Values and error bars indicate the mean ± SD.

**Figure 6 molecules-28-03079-f006:**
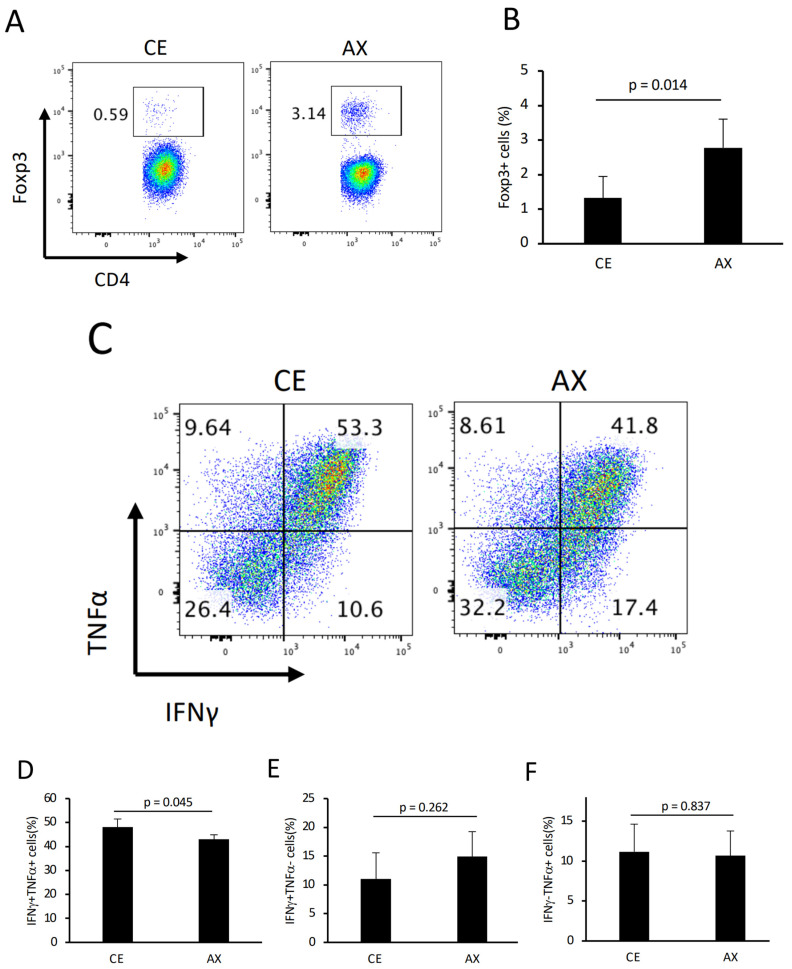
The effect of AX on colonic T cells in colitis model mice: (**A**) Typical FACS panel showing the percentage of Foxp3^+^ T cells 6 weeks after the adoptive transfer of naïve CD4^+^ T cells in the Rag1^−/−^ mice. (**B**) Percentage of Foxp3^+^ T cells gated on CD4+. (**C**) Typical FACS panel showing the IFNγ− and/or TNFα-producing CD4+ T cells. (**D**–**F**). The percentage of IFNγ^+^/TNFα^+^, IFNγ^+^/TNFα^−^, and IFNγ^−^/TNFα^+^ T cells gated on CD4+. Values and error bars indicate the mean ± SD.

## Data Availability

Not applicable.

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
