# Peer review of "Effect of Wheat-Derived Arabinoxylan on the Gut Microbiota Composition and Colonic Regulatory T Cells"

_molecules, 2023, doi:10.3390/molecules28073079_

Round 1

Reviewer 1 Report

In this study submitted to molecules, the authors assessed the effects of wheat-derived arabinoxylan (AX) on gut microenvironment.

There was until this work an inconsistency between the probiotic effects of AX observed in vitro and in vivo under basal conditions, and the only reference assessing its effects under IBD conditions, showing no effects of AX on DSS-induced colitis. The authors demonstrated that AX disturbed gut microbiota composition (notably Firmicutes, Bacteroides, Proteobacteria phyla, Lactobacillus and Lachnospiraceae species), increased propionate and butyrate SCFA levels, and increased colonic T reg population. They also showed that AX prevented chronic colitis induced by T cell transfer. The methodology was well chosen. All the parts of this paper are clear, well organized, with accurate references, sufficient details, no misinterpretation. The discussion is well written.

However, I have few minor comments aiming to improve the manuscript.

The duration of AX supplementation is lacking in several part. In the methods 4.1., in the results 2.2.to 2.4.

The statement “for more than 2 weeks” in the abstract should be more precise.

Line 33 Please correct Crohn’s disease.

Line 36. The reference 2 is a little old. Is it still true currently?

Line 163 Please define p of pTreg cells.

Please revise lines 191 to 195 because there is an inconsistency with the figure and please be cautious because the decrease of IFNg+TNFa+ cells is the only significant result.

Line 279. It would be useful to estimate the concentration of AX in mg/kg of food as well as in mg/kg bw.

Author Response

The duration of AX supplementation is lacking in several part. In the methods 4.1., in the results 2.2.to 2.4.

We apologize that the duration of AX feeding was missing. We added this information to the method and result sections (Lines 98, 151, and 168 in the revised manuscript)

The statement “for more than 2 weeks” in the abstract should be more precise.

We changed this sentence (Lines 18–19 in the revised manuscript).

“In this study, healthy and chronic colitis model mice were fed chow containing cellulose or wheat-derived arabinoxylan for 2–6 weeks and subjected to subsequent analysis.”

Line 33 Please correct Crohn’s disease.

We have corrected this typo in the revised manuscript (Line 32 in the revised manuscript). Thank you for your indication.

Line 36. The reference 2 is a little old. Is it still true currently?

To our knowledge, this guideline remains the standard and was also cited in a recent review describing the role of dietary fiber in the management of IBD (we’ll be glad if you refer to the following review article: https://doi.org/10.3390/nu14224775).

Line 163 Please define p of pTreg cells.

The first use of abbreviated “pTreg” was in line 88 in the previous introduction (Line 91 in the revised manuscript), which was before Line 163.

Please revise lines 191 to 195 because there is an inconsistency with the figure and please be cautious because the decrease of IFNg+TNFa+ cells is the only significant result.

We apologize for our excessive interpretation in the result section. We changed this sentence (Lines 198–200 in the revised manuscript).

“In this Th1-prone model, AX intervention significantly decreased IFNg/TNFa double positive cells but not IFNg or TNFa single positive cells compared to the CE group (Fig. 6C–F).”

 Corresponding to this change, we deleted the following sentence from the previous version of the manuscript.

“As a result, IFNg-producing T cells were comparable between the CE- and AX-fed mice but TNFa-producing T cells were less abundant in the AX group.”

Line 279. It would be useful to estimate the concentration of AX in mg/kg of food as well as in mg/kg bw.

The estimated concentration of AX in mg/kg food and that in mg/kg bw were as follows:

mg/kg food: 50 mg/kg

mg/kg body weight per day : 600 mg/kg

We added this information to the revised material and methods section (Lines 285–287)

Reviewer 2 Report

The authors have reported good evidence for the mouse gut microbial composition-modifying effects of wheat arabinoxylan and its immune-stimulation in a T cell transfer model of chronic colitis. It would be very helpful to know more about the wheat arabinoxylan composition (degree of polymerization and degree of substitution) in order to compare these results with the literature. Other minor suggestions for consideration are listed below.

p. 1, line 33: Crohn’s disease

p. 1, line 37: Dietary fiber is a mixture of polysaccharides that include 10 or more monosaccharides …

p. 2, line 48: … are dependent of their solubility, molecular …

p. 2, line 49: … viscosity, hydrophilicity, degree of branching, monosaccharide content, …

p. 2, lines 51-53: Explain why arabinoxylan is a highly fermentable fiber in one sentence and then a slowly fermenting fiber in the next sentence. I think you mean that arabinoxylan is highly fermentable but that fermentation takes place in the transverse to distal colon locations rather than the proximal colon.

p. 2, lines 54-55: Cellulose is known to not be fermented in the human gastrointestinal tract due to the lack of cellulolytic microbes. If reference 14 reported that human microbiota could degrade cellulose in vitro, then how do we know that will happen in the human gut?

p. 3 and 4: Lactobacilliaceae and Lachnospiraceae were identified as elevated in arabinoxylan-fed mice yet why was only Lachnospiraceae listed in the abstract?

Author Response

The authors have reported good evidence for the mouse gut microbial composition-modifying effects of wheat arabinoxylan and its immune-stimulation in a T cell transfer model of chronic colitis. It would be very helpful to know more about the wheat arabinoxylan composition (degree of polymerization and degree of substitution) in order to compare these results with the literature. Other minor suggestions for consideration are listed below.

Thank you for your suggestion regarding wheat arabinoxylan’s structure. We added the following sentences to the introduction (Lines 51–56 in the revised manuscript).

“In general, AX consists of linear b(1-4) linked xylan backbone to which a-1-arabinofuranose units are attached as side residues via a-(1-3) and/or a(1-2) linkage. Xyloses are most commonly mono-substituted but the degree of substitution is influenced by the wheat variety and wheat grain’s maturation. Compared with rice bran AX, wheat AX has a less branched structure, and (1-5)-linked arabinose is absent (reviewed in [13]).”

  1. 1, line 33: Crohn’s disease

We have corrected this typo in the revised manuscript. Thank you for your indication.

  1. 1, line 37: Dietary fiber is a mixture of polysaccharides that include 10 or more monosaccharides …
  2. 2, line 48: … are dependent of their solubility, molecular …
  3. 2, line 49: … viscosity, hydrophilicity, degree of branching, monosaccharide content, …

We appreciate the reviewer’s helpful suggestion. We have corrected these sentences accordingly (Lines 36, 47, and 48 in the revised manuscript).

  1. 2, lines 51-53: Explain why arabinoxylan is a highly fermentable fiber in one sentence and then a slowly fermenting fiber in the next sentence. I think you mean that arabinoxylan is highly fermentable but that fermentation takes place in the transverse to distal colon locations rather than the proximal colon.

We appreciate the reviewer’s helpful suggestion. We changed this sentence (Lines 56–58 in the revised manuscript).

Among the many kinds of dietary fibers, arabinoxylan (AX) derived from wheat is known to be a highly fermentable fiber owing to its relatively high degree of arabinose substitution on the xylan backbone [12]. ~~~. Wheat AX fermentation takes place in the transverse to distal colon locations rather than the proximal colon, resulting in slow fermentation and effective alteration in the gut microenvironment [14].

  1. 2, lines 54-55: Cellulose is known to not be fermented in the human gastrointestinal tract due to the lack of cellulolytic microbes. If reference 14 reported that human microbiota could degrade cellulose in vitro, then how do we know that will happen in the human gut?

As the reviewer indicated, cellulose is poorly fermented by the gut microbiota of non-ruminant mammals including humans and rodents. On the contrary, cellulose intervention has been recently reported to contribute to a change in gut microbiota composition (Kim, et al. Gut Microbes 2020. https://doi.org/10.1080/19490976.2020.1730149).

It might be a result of the digestion (but not fermentation) of cellulose by the gut microbiota because reference #14 reports that the human microbiota can degrade (not ferment) cellulose in vitro. Therefore, we wrote that cellulose can be degraded (but not fermented) by the human gut microbiota.

However, we don’t know the precise method to detect cellulose degradation in the human gut in vivo. In addition, cellulose degradation by human gut microbiota is not a pivotal issue in our manuscript.

Therefore, we deleted text on cellulose degradation by human gut microbiota and only described that cellulose is poorly fermented in the human gut (Line 60 in the revised manuscript).

[Previous]

In contrast, fermentation of cellulose (CE), the most abundant dietary fiber in plants, is minimal, although it can be degraded by human gut microbiota [14].

[Revised]

In contrast, fermentation of cellulose (CE), the most abundant dietary fiber in plants, is minimal in the human gut [15].

  1. 3 and 4: Lactobacilliaceae and Lachnospiraceae were identified as elevated in arabinoxylan-fed mice yet why was only Lachnospiraceae listed in the abstract?

In this study, we particularly focused on Lachnospiraceae as AX-responsive bacteria because this family reportedly contributes to butyrate production and Treg induction. Lactobacillus spp. are well-characterized in affecting host health; however, to our knowledge, Lactobacillus spp.’s involvement in Treg induction through butyrate production has not been reported. The health benefits of AX through elevated levels of Lactobacillus spp. are intriguing but beyond the scope of this manuscript. Therefore, we described Lachnospiraceae as AX-responsive bacteria in the abstract.